# Association between Self-Reported Prior Night’s Sleep and Single-Task Gait in Healthy, Young Adults: A Study Using Machine Learning

**DOI:** 10.3390/s22197406

**Published:** 2022-09-29

**Authors:** Ali Boolani, Joel Martin, Haikun Huang, Lap-Fai Yu, Maggie Stark, Zachary Grin, Marissa Roy, Chelsea Yager, Seema Teymouri, Dylan Bradley, Rebecca Martin, George Fulk, Rumit Singh Kakar

**Affiliations:** 1Honors Program, Clarkson University, Potsdam, NY 13699, USA; 2Sports Medicine Assessment Research & Testing (SMART) Laboratory, George Mason University, Manassas, VA 20110, USA; 3Department of Computer Science, George Mason University, Manassas, VA 20110, USA; 4Department of Medicine, Lake Erie College of Osteopathic Medicine, Elmira, NY 14901, USA; 5Latham Medical Group, Latham, NY 12110, USA; 6Department of Engineering and Technology, State University of New York Canton, Canton, NY 13617, USA; 7Department of Physical Therapy, Hanover College, Hanover, IN 47243, USA; 8Department of Neurology, St. Joseph’s Hospital Health Center, Syracuse, NY 13203, USA; 9Department of Physical Therapy, Emory University School of Medicine, Atlanta, GA 30322, USA; 10Human Movement Science Department, Oakland University, Rochester, MI 48309, USA

**Keywords:** partial sleep deprivation, sleep extension, lower extremity kinematics, gait assessment

## Abstract

Failure to obtain the recommended 7–9 h of sleep has been associated with injuries in youth and adults. However, most research on the influence of prior night’s sleep and gait has been conducted on older adults and clinical populations. Therefore, the objective of this study was to identify individuals who experience partial sleep deprivation and/or sleep extension the prior night using single task gait. Participants (n = 123, age 24.3 ± 4.0 years; 65% female) agreed to participate in this study. Self-reported sleep duration of the night prior to testing was collected. Gait data was collected with inertial sensors during a 2 min walk test. Group differences (<7 h and >9 h, poor sleepers; 7–9 h, good sleepers) in gait characteristics were assessed using machine learning and a post-hoc ANCOVA. Results indicated a correlation (r = 0.79) between gait parameters and prior night’s sleep. The most accurate machine learning model was a Random Forest Classifier using the top 9 features, which had a mean accuracy of 65.03%. Our findings suggest that good sleepers had more asymmetrical gait patterns and were better at maintaining gait speed than poor sleepers. Further research with larger subject sizes is needed to develop more accurate machine learning models to identify prior night’s sleep using single-task gait.

## 1. Introduction

Many adults do not meet the minimum recommended 7–9 h of sleep per night [1,2] and as a result experience sleep deprivation (SD) [3]. Common causes of SD are health issues [4], work schedules [5], travel [6] and serving as a caregiver to a family member [7]. SD can be classified as acute partial SD (APSD), acute total SD and chronical partial SD. APSD is defined as 4–6 h per night whereas acute total SD is characterized by extended wakefulness for 24–72 h [8,9]. APSD is common for a variety of reasons and most individuals are likely to experience APSD on a regular basis [8]. Often individuals experiencing SD are otherwise healthy, young adults. For example, tactical populations [10] and athletes [11] routinely have bouts of SD due to their schedules. One common problem for both these populations are high rates of injury [12] which have been linked to sleep in several reviews [13,14]. There is evidence that SD impairs cognitive function [15] and subsequently psychomotor performance of motor tasks [8,9]. Perhaps one of the most common motor tasks performed daily is gait which is regulated by both automatic and executive control processes [16]. Gait characteristics in the trunk and lower extremity have been found to be associated with greater injury risk [17] as well as prior injury [18]. Furthermore, greater variability in gait patterns may indicate impairments to neurophysiological function [19] that could increase injury or fall risk [20]. Assuming that gait variability is regulated by automatic processes an argument could be made that when cognitive function is compromised an increase in gait variability would be observed due to a consequent impairment of automaticity. Thus, it is logical that cognitive impairments due to SD may lead to altered gait patterns.

Several prior studies provide evidence that SD affects several features of gait and balance [21,22,23,24,25,26]. Most notably, a recent 2018 study by Howell and colleagues found that self-reported sleep duration is associated with gait during a tandem gait task [23] but not steady state gait in healthy collegiate athletes. The researchers utilized a cutoff of 7 h of sleep during their analysis based on sleep duration recommendations. While no significant effects of sleep on steady-state gait were observed, the authors did find those with <7 h had changes in during tandem gait which included significantly longer double support times in walking only (i.e., single-task) and walking while simultaneously performing a cognitive task (i.e., dual-task) conditions [23]. Agmon et al. reported that in older adults lower sleep efficiency is related to decreased gait speed whereas longer sleep latency is related to increased stride-variability [21]. Agmon and colleagues concluded that abnormal sleep behavior is related to increased overall gait variability [21]. The influence of chronic sleep quality on gait has been studied in healthy, young adults as Liu and colleagues (2019) reported that sleep quality was associated with gait [22]. Their findings showed that sleep quality had a greater influence on the upper body rather than lower extremity gait characteristics [22]; however, they did not present gait parameters (i.e., gait speed, step length, stride rate, limb asymmetry) normally examined in gait literature [27]. This brings into question whether sleep influenced gait or the posture of participants which would have been manifested in gait measures reported by Liu and colleagues [22]. Acute changes in sleep duration have been shown to affect postural control in healthy individuals [24,25,28,29]. Similar to gait, postural control is regulated by both automatic and executive control processes [16]. Thus, supraspinal areas [30,31] impacted by SD would likely have an effect on control of both gait and balance [24,25,28,29]. A 2021 study by Umemura and colleagues compared the effects of acute to chronic SD on gait control [26]. Interestingly, while both acute and chronic SD led to worse performance, chronic SD performed better than the acute SD. The finding indicates that individuals may adapt to SD over time.

While existing research investigating the relationship of sleep with gait focuses on SD, sleep extension (SE), has been shown to have an effect on neurocognitive, measures, alertness and vigilance [32,33]. SE is defined as sleeping maximally or >9 h [32]. A number of studies on collegiate and elite athletes show that SE improves motor control, accuracy, and fine motor coordination [34,35,36]. Several recent studies have reported that excessive sleep duration is associated with cognitive decline [37] as well as chronic disease, worse quality of life and bad physical function [38]. Therefore, it is plausible that gait could be affected by SE. However, there is evidence that more sleep is not necessarily better as extended sleep durations have been associated with negative health effects [1]. To date the effects of SE on gait has not been studied. 

There are several limitations of the current literature regarding the effect of SD on gait characteristics in healthy adults which present opportunities to further knowledge in this area of study. For example, literature regarding the effects of acute partial SD on single-task gait in young, healthy adults is scarce and has not been replicated [23]. Additionally, we are unaware of any studies that have examined gait differences in extended sleepers (>9 h) and individuals who receive 7–9 h of sleep the night prior. Sleep extension (SE), has been shown to have an effect on neurocognitive measures, alertness and vigilance [32,33]. Several studies on collegiate and elite athletes show that SE improves motor control, accuracy, and fine motor coordination [34,35,36]. Therefore, it is plausible that gait could be affected by SE. We are also unaware of any studies that have used machine learning algorithms to identify individuals who report APSD and/or extended sleep from those who report 7–9 h of sleep. However, a recent study by Kokkotis and colleagues indicated that machine learning approaches may be able to identify alterations to gait which are neglected by traditional statistical methods [18]. Therefore, the purpose of this study was to use machine learning algorithms to identify individuals who report APSD, SE and those who report 7–9 h of sleep. 

## 2. Materials and Methods

### 2.1. Participants

There were 123 participants between the ages of 18 and 36 in the study (males = 46, females = 77; Table 1). Healthy subjects were recruited from a university population of undergraduate and graduate students using a combination of in-class announcements, email lists, and flyers on campus. The university is primarily an engineering institution located in a small town in Upstate New York. For the purposes of this study healthy was defined as individuals not presenting with musculoskeletal pain or injury in the past six months and who had the ability to walk for extended periods with no difficulty. Specific inclusion and exclusion criteria required subjects to be able to walk unassisted for two minutes and not have any lower extremity functional impairment or have pain or discomfort while walking, any neurological conditions (i.e., stroke), or lower extremity orthopedic surgery within the last six months. The study was approved by the institutional review board of Clarkson University (approval #18.39.1) and prior to participating subjects signing an informed consent.

### 2.2. Experimental Procedure

Participants completed the protocol during a single session lasting approximately 75 min. Testing occurred between 9 am and noon for all participants to minimize effects of time of day on the results [39]. After completing the informed consent, height and weight were measured. Participants’ height was measured using a stadiometer (SECA model 220, SECA Corporation, Chino, CA, USA). Weight was measured using the Tanita Bioelectrical Impedance Analysis Scale (TBF-410, Tanita Corporation, Tokyo, Japan).

Participants completed a set of open-ended questions about their prior night’s sleep. This survey collected information pertaining to the time the participant went to bed, the time they turned the lights out, how much time it took them to fall asleep once in bed, the number of times they woke up during the night, the total amount of time (in minutes) they were awake throughout the night, and the time they woke up. The total hours of sleep were calculated by finding the total amount of time a participant was in bed (when they went to bed until they woke up) and subtracting the time that it took the participant to fall asleep as well as the amount of time they were awake during the night. The amount of times a participant woke up in the night was marked as a sleep disturbance [40].

Gait data was collected using inertial sensors (APDM’s Mobility Lab^TM^, APDM Inc., Portland, OR, USA) during a two-minute walk test. The inertial sensors were attached to the body in seven locations (sternum, lower back, forehead, left foot, right foot, left wrist, and right wrist) using Velcro^TM^ straps. The validity and reliability of the APMD’s ability to measure gait has previously been established [41]. The two-minute walk test was performed around a 6 m indoor pathway marked by two orange cones at a self-selected pace and has been used previously [42,43]. Notably this task required subjects to frequently make left turns and gait patterns during turning have been shown to be more sensitive to impairments in clinical populations [44] as well as age-related changes in gait development [45]. Gait characteristics were sampled at 128 Hz and processed using APDM’s Mobility Lab before being exported [41]. Descriptions of all the gait characteristics are provided in a Appendix A. We indicate in the table which characteristics were collected during ambulation along a straight path versus while the participant was turning. However, readers should note that upper body characteristics have limited ability to discriminate between gait in turns and straight-away [46].

### 2.3. Statistical Analysis

#### 2.3.1. Data Pre-Processing

Data for demographics, prior night sleep duration, and gait were initially compiled into a Microsoft Excel (Microsoft Inc., Redmond, WA, USA) database. Data for each gait variable and prior night’s sleep were visualized together. Our data visualization yielded a U-shaped pattern in the data with individuals reporting <7 h of sleep displaying similar gait characteristics as those reporting >9 h of sleep. Therefore, we created 2 categories of data, good sleepers (7–9 h of sleep) and poor sleepers (<7 h or >9 h of sleep) based on whether participants obtained the recommended sleep duration the prior night. Data was then processed in Python (version 3.8.5, Python Software Foundation, Wilmington, DE, USA). During data cleaning it was observed that some data points were not detected by the equipment. This resulted in missing data points for some participant data. In the event more than 5% of features (i.e., gait variable) were missing for a participant the feature was removed from the data set. In cases where less than 5% of features were missing we then filled in missing values through the mean value of that feature in the dataset [47]. After pre-processing we had 123 participants with 56 valid features.

#### 2.3.2. Machine Learning Models

When recording high dimensional features not every feature is equally important, and there may be redundant features that are of less importance. Therefore, to sort through features we used the Random Forest according to their importance [48]. After sorting the features, we used the dataset to train the model through Regressors and Classifiers respectively. For the Classifier, we classified the records as poor sleepers (<7 h or >9 h of sleep) and the rest as good sleepers. We used all features and top 9 features (using 0.03 as a cut-off for feature importance) to train each model. We trained the models in a 10-fold cross-validation manner in order to avoid problems such as overfitting or selection bias [49]. We used mean absolute error to assess the Regressors and “Accuracy” to evaluate Classifier models. We also assessed correlation coefficients (R2) between the predicted prior night sleep duration and the self-reported prior night sleep duration on the Regressor Models [22]. Using a Monte Carlo method, we randomly split the training set (90%) and test set (10%) and ran each of the ML models 10,000 times.

#### 2.3.3. Post-Hoc Analysis

A post-hoc analysis of co-variance (ANCOVA) was used to determine statistically significant differences between the poor sleeper and good sleeper groups for gait features. The ANCOVA accounted for sex [50], age [51,52], height [52], and weight [52,53] as these factors have been shown to influence gait.

## 3. Results

There were 59 participants (48%) who reported <7 h or >9 h and 64 with 7–9 h of sleep the previous night (Table 1). There were no significant differences in height or mass (*p* > 0.05) between the two groups; however, older individuals had a statistically greater chance of reporting poor sleep (*p* = 0.003).

### 3.1. Feature Importance

The most important feature was variance in the toe out angle (relative importance = 6.2%), which is 0.59% higher than the second place (foot strike angle). The least important variable is the characteristic variance in gait cycle duration. All correlation coefficients for the top 9 features were <0.65 suggesting that the top 9 features were relatively independent. Characteristics with their importance and their means for each category can be found in Table 2.

### 3.2. Model Evaluation

Overall, the best performance of the Classifier was achieved by Random Forest on the top 9 features; the median of the accuracy was 61.54%, and the mean accuracy was 65.03%. The highest accuracy for all models was 100% (0.3% of all models run) (Table 3).

### 3.3. Post-Hoc

Post-hoc ANCOVA for the top 9 features revealed that poor sleepers had less variance in their toe out angle and smaller trunk angles. However, poor sleepers demonstrated larger foot strike angles, greater frontal plane bending, and increased variance in cadence. Of the other variables included in our models, poor sleepers had longer stride lengths, more time spent in single leg support time, and swing phase time, took more time in turns, and had greater transverse plane range of motion. Good sleepers had greater variance in foot strike angles and spent more time in double leg support time.

To illustrate differences in those who obtained 7–9 h of sleep versus the rest of the subjects, a computer model was created based on the data from the study and is available on the website (https://gaitsim.dmanserver.com/Sleep, accessed on 18 September 2022).

## 4. Discussion

To our knowledge this is the first study to use machine learning to identify gait characteristics in individuals who obtained the recommended amount of sleep, were partially SD, or who experienced SE (sleep extension). The findings of this study suggest that those who obtain 7–9 h of sleep the prior night exhibit a different single-task gait pattern than those who slept less or more. Our main findings and post-hoc analyses taken together suggest that those who self-reported 7–9 h of sleep had more asymmetrical gait with slower gait speed and less trunk motion as compared to those who reported acute SD or SE. However, those who reported APSD or SE had trouble maintaining gait speed as evidenced by increased variance in cadence and larger stride lengths and less time spent in single leg support time as compared to those who received the recommended amount of sleep.

The results from our primary and post-hoc analyses suggest that individuals who report getting more or less than the recommended amount of sleep had gait patterns consistent with those who are trying to ambulate faster but are unable to maintain a steady gait speed as evidenced by increased variance in cadence, gait speed, and circumduction by individuals who report more or less than the recommended amount of sleep. While our post-hoc analysis only reported significant findings for variance in cadence, machine learning allowed us to gain a better understanding of the variance in gait for poor sleepers. Machine learning identified that lower extremity variances and frontal plane bending were all predictors of prior night’s sleep (top 9 features). Poor sleepers in general demonstrated gait patterns that were consistent with individuals trying to maintain gait speed (see relative importance column of Table 2). Good sleepers conversely maintained gait speed through a more efficient gait pattern.

Howell et al. [23] reported no significant differences in any of the gait parameters for single and dual-task steady-state gait performed at a self-selected pace, whereas our study did find subtle differences in several individual gait parameters (Table 2). An important difference between our study and that of Howell et al. is that their single-task gait task was to walk to a marker 10 m away then return to the starting position whereas our task was 2-min of continuous gait along a track. While the gait task chosen by Howell et al. created an increased external attentional focus which may decrease the impact of supraspinal input on gait [54,55,56], the two-step command while ambulating in a new and clinical setting may have altered the available resources of the executive control [57]. As stated in the conclusion by Howell et al. [23], it has been found that SD has a significant effect on areas of the brain [31] likely involved with attention and motor performance [58]. This may also explain their lack of significant findings [23].

While Howell and colleagues [23] reported no effect of sleep duration on single or dual-task gait, a study by Agmon et al. [21] reported that lower sleep efficiency is associated with decreased gait speed and increased variability in dual-task gait in an older adult population [21]. Due to differences in the population studied as well as measures of sleep by Agmon et al. [21] directly comparison with our findings should be made with caution. Thus, while a dual-task condition was not included in our protocol, we would postulate that, based on the findings of Howell et al. [23], similar differences in gait between sleep groups would have been observed as we reported for single-task gait. This study was part of a larger data collection process, and a dual-task gait condition was not included due to the length of the study protocol for each participant. Admittedly this is a limitation and future studies should include both single- and dual-task gait conditions.

An interesting finding in our data was the similarities in gait between those who reported SD compared to those who reported SE. To our knowledge the effects of SE on gait have not been reported previously. Previous studies have reported that SE can result in improved physical performance on skilled tasks [35,36]. Through graphical representations and pre-processing of our data we identified no differences between the gait characteristics of SD and SE. Although not reported in this manuscript, we conducted analyses that found significant differences between gait variables for SE vs normal sleepers, but no significant differences between SE and SD (see Appendix A). Our findings suggest the relationship between sleep duration and gait performance may be characterized by a ‘U’ relationship. This should be explored further in the future.

Additionally, we find that our study had high correlation between gait characteristics and sleep duration (79%); however, we had poor classification accuracy (~65%). We should note that approximately 0.3% of all models had 100% classification accuracy. These results suggest that there is a relationship between single-task gait and self-reported sleep duration. However, significantly more subjects are needed to be able to accurately predict prior night’s sleep from single-task gait in this population. The models that had 100% accuracy were “lucky” in that the participants who were randomly selected for their training and test models and that these algorithms were perhaps over-fitting.

As is the case with any study, there are several limitations. Self-reported sleep data has limited validity [59] and sleep needs have high inter-individual variability [60]. More specifically, correlations between self-reported and objectively measured sleep durations have been reported to have a moderate relationship and individuals sleeping <7 h tend to over report actual sleep duration [59]. We also did not account for the reasons that may have caused SE or SD (i.e., sleeping in due to prior sleep loss, short sleep or alcohol use the night before). Additionally, we did not screen participants for mental disorders, such as depression, or pharmacological substances which have been reported to affect gait [61]. Our approach to dichotomous participnats into good and poor sleepers based on population level sleep duration recommendations [1,2] may be another limitation. Given a growing body of literature supporting a strong genetic component in terms of individual sleep needs [62], where some individuals may function equally as well to others on less sleep, a more senstive approach to inter-individual differences would be advised in the future. A potential limitation of the classification accuracy of the machine learning models is that the gait and sleep time data had a parabolic relationship, suggesting that someone who slept 6.9 h had gait parameters that were very similar to those of someone who slept 7.0 h. This might be a reason for the high correlation and poor classification accuracy of our models. However, since the objective of this study was to identify individuals who met sleep requirements compared to those who did not, this study did not conduct additional analyses that could predict parabolic data. Future research should include objective measures of sleep duration and investigate the effects of differences in acute sleep deprivation from typical night sleep duration on gait to address these shortcomings.

## 5. Conclusions

The objective of this study was to determine differences in single-task gait characteristics between those who report SD, SE, and individuals who get the normal amount of sleep. The findings from our study suggest that gait differences exist between poor and good sleepers, but none between poor sleepers who experience SD vs SE. Additionally, our machine learning models were able to classify within 65% accuracy those who self-reported poor sleep (SD and SE) and those who self-reported good sleep. We found that those who report more or less than the recommended amount of sleep have a more difficult time maintaining gait speed. A stronger effect of sleep on gait would be expected in clinical populations based on prior literature [16,21,31] or if dual-task gait is performed [21]. For clinicians or researchers assessing gait we recommend asking individuals about the duration of their prior night’s sleep as some individuals may have sleep-related gait impairments. When individuals report SD or SE, increased stride length or transverse plane range of motion or increased variability of cadence may be related to sleep during the prior night, rather than a pathological gait related to the health condition of the individual.

## Figures and Tables

**Table 1 sensors-22-07406-t001:** Participant characteristics.

	Height (cm)	Weight (kg)	Age (years)	Sex (Male: Female)
Good Sleepers (n = 64)	173.39	74.25	23.56	19:45
Poor Sleepers (n = 59)	173.19	74.12	24.90	27:32
test statistic/*p*-value	0.84/0.87	0.95/0.95	0.003 **/0.02 *	N/A

Good sleepers = 7–9 h of sleep; Poor Sleepers = <7 or >9 h of sleep; * *p* < 0.05, ** *p* < 0.01.

**Table 2 sensors-22-07406-t002:** Feature importance and descriptive statistics.

			Good Sleepers (n = 64)		Poor Sleepers (n = 59)			
Feature	Relative Importance	Ranking	Mean	SD	Mean	SD	Sig. Diff.?	Finding
Toe Out Angle Variance (%)	6.19%	1	2.79	1.95	1.93	1.36	Yes	Good > Poor
Foot Strike Angle (deg)	5.60%	2	23.21	4.54	25.26	4.09	Yes	Poor > Good
Back Right Frontal Plane Bending Angle (deg)	5.58%	3	2.58	2.32	3.75	2.26	Yes	Poor > Good
Cadence Variance (%)	4.94%	4	0.18	0.17	0.29	0.22	Yes	Poor > Good
Trunk Angle (deg)	3.83%	5	187.29	4.69	186.12	3.71	Yes	Good > Poor
Terminal Double Leg Support Variance (%)	3.63%	6	3.74	3.35	3.20	3.72		
Gait Speed Variance (%)	3.32%	7	0.94	0.83	1.10	0.64		
Circumduction Variance (%)	3.19%	8	18.14	14.00	18.47	13.63		
Toe Out Angle (deg)	3.15%	9	37.15	2.94	36.65	3.61		
Double Leg Support Variance (%)	2.45%	10	0.60	0.44	0.69	0.59		
Stride Length (m)	2.41%	11	1.18	0.09	1.22	0.12	Yes	Poor > Good
Foot Strike Angle Variance (%)	2.17%	12	4.39	3.91	3.49	2.98	Yes	Good > Poor
Single Leg Support Variance (%)	2.16%	13	0.90	0.83	1.02	0.83		
Trunk Frontal Plane ROM (deg)	2.08%	14	4.64	1.79	5.14	2.01		
Trunk Transverse Plane ROM (deg)	2.08%	15	10.31	3.76	11.58	2.92	Yes	Poor > Good
Back Frontal Plane ROM (deg)	2.00%	16	8.69	2.55	9.50	3.14		
Back Left Frontal Plane Bending Angle (deg)	1.97%	17	6.11	2.49	5.75	2.83		
Arm Swing Velocity (deg/s)	1.84%	18	190.34	71.10	192.4	62.69		
Stance (% Gait Cycle)	1.81%	19	60.54	1.52	59.87	1.54	Yes	Good > Poor
Toe Out Angle (deg)	1.80%	20	4.74	5.26	4.22	6.71		
Arm Swing Velocity Variance (%)	1.67%	21	11.26	8.63	11.04	9.39		
Trunk Transverse ROM (deg)	1.63%	22	8.04	2.89	8.44	2.36		
Steps in Turn (#)	1.59%	23	3.53	0.31	3.52	0.34		
Back Sagittal Plane Minimum Angle (deg)	1.50%	24	−2.03	4.38	−2.23	5.52		
Turns Duration (s)	1.45%	25	2.19	0.21	2.21	0.24		
Single Leg Support (%GCT)	1.41%	26	39.43	1.51	40.03	1.48	Yes	Poor > Good
Trunk Transverse Plane ROM (deg)	1.40%	27	9.33	2.57	9.16	2.44		
Back Sagittal Plane ROM (deg)	1.27%	28	6.10	2.24	6.14	1.98		
Double Leg Support (% Gait Cycle)	1.25%	29	21.11	3.02	19.84	3.01	Yes	Good > Poor
Step Variability	1.24%	30	2.82	0.67	2.95	0.73		
Back Transverse Plane Left Rotation Maximum Angle (deg)	1.20%	31	3.33	12.61	2.91	14.54		
Step Variability Variance (%)	1.18%	32	10.82	7.66	10.33	7.20		
Arm ROM Variance (%)	1.16%	33	13.37	10.71	14.24	11.83		
Mid-swing Elevation Variance (%)	1.09%	34	17.58	16.47	16.63	12.70		
Circumduction (cm)	1.07%	35	2.85	1.10	2.87	1.20		
Mid-swing Elevation (cm)	1.07%	36	1.29	0.62	1.32	0.65		
Back Transverse Plane Right Rotation Maximum Angle (deg)	1.06%	37	6.98	13.26	8.66	14.16		
Stance Variance (%)	1.05%	38	0.58	0.55	0.66	0.56		
Swing Phase (% Gait Cycle)	1.04%	39	39.46	1.52	40.13	1.54	Yes	Poor > Good
Lumbar Sagittal Plane ROM (deg)	1.04%	40	5.42	1.41	5.38	1.66		
Gait Speed (m/s)	0.95%	41	1.04	0.13	1.07	0.14		
Swing Variance (%)	0.92%	42	0.88	0.84	0.99	0.85		
Lumbar Frontal ROM (deg)	0.92%	43	5.69	1.98	6.28	2.31		
Terminal Double Leg Support (%GCT)	0.88%	44	10.61	1.49	9.99	1.50	Yes	Good > Poor
Trunk Sagittal Plane ROM (deg)	0.77%	45	5.49	1.61	5.48	1.56		
Arm ROM (deg)	0.76%	46	41.99	16.29	43.31	15.05		
Back Sagittal Plane Maximum Angle (deg)	0.76%	47	4.07	4.21	3.91	5.42		
Step Duration Variance (%)	0.74%	48	1.02	0.92	0.96	0.84		
Turn Velocity (deg/s)	0.70%	49	179.64	25.56	184.50	31.11		
Cadence (step/min)	0.65%	50	105.36	8.23	105.66	8.67		
Stride Length Variance (%)	0.58%	51	0.89	0.69	0.93	0.61		
# of Turns	0.55%	52	16.71	2.26	17.32	2.29	Yes	Poor > Good
Gait Cycle Duration (s)	0.39%	53	1.15	0.09	1.14	0.09		
Step Duration (s)	0.35%	54	0.57	0.04	0.57	0.05		
Gait Cycle Duration Variance (%)	0.17%	55	0.17	0.23	0.23	0.29		

1. Abbreviations: deg, degrees; GCT, ground contact time; ROM, range of motion. 2. Variance was computed at the % difference between left and right sides. 3. Differences between good and bad sleepers were tested with independent samples *t*-tests. 4. Trunk and upper extremity variables are shaded light gray. Lower extremity kinematic and gait variance variables are shaded dark gray.

**Table 3 sensors-22-07406-t003:** Model evaluation results.

Regressors R^2^	Rank	Mean	SD	Minimum	25%	50%	75%	Maximum
Random Forest Top 9	1	−0.53	0.67	−7.67	−0.86	−0.55	−0.08	1.00
Ada Boost Top 9	2	−0.63	0.67	−7.67	−0.95	−0.55	−0.24	1.00
Random Forest Full	3	−1.01	0.71	−10.92	−1.36	−0.95	−0.55	1.00
Support Vector Class Top 9	4	−1.34	1.07	−12.00	−1.60	−1.17	−0.86	1.00
Ada Boost Full	5	−1.10	0.69	−9.83	−1.48	−1.17	−0.63	0.69
Support Vector Class Full	6	−1.47	0.99	−12.00	−1.81	−1.17	−0.86	0.03
**Regressors Mean Absolute Error**	**Rank**	**Mean**	**SD**	**Minimum**	**25%**	**50%**	**75%**	**Maximum**
Random Forest Top 9	1	0.35	0.13	0.00	0.23	0.31	0.46	0.77
Ada Boost Top 9	2	0.37	0.13	0.00	0.31	0.38	0.46	0.85
Random Forest Full	3	0.46	0.13	0.00	0.38	0.46	0.54	0.92
Ada Boost Full	4	0.48	0.13	0.08	0.38	0.46	0.54	1.00
Support Vector Class Top 9	5	0.52	0.13	0.00	0.46	0.54	0.62	1.00
Support Vector Class Full	6	0.55	0.12	0.23	0.46	0.54	0.62	1.00
**Classifiers**	**Rank**	**Mean**	**SD**	**Minimum**	**25%**	**50%**	**75%**	**Maximum**
Random Forest Top 9	1	65.03%	12.67%	15.38%	53.85%	61.54%	76.92%	100.00%
Ada Boost Top 9	2	62.71%	13.30%	15.38%	53.85%	61.54%	69.23%	100.00%
Random Forest Full	4	54.26%	13.12%	7.69%	46.15%	53.85%	61.54%	100.00%
Ada Boost Full	3	52.20%	13.02%	7.69%	46.15%	53.85%	61.54%	100.00%
Support Vector Class Top 9	5	47.72%	12.75%	0.00%	38.46%	46.15%	53.85%	92.31%
Support Vector Class Full	6	45.06%	12.28%	0.00%	38.46%	46.15%	53.85%	61.54%

Models were evaluated using a Monte Carlo method and data were randomly split into 90% training set and 10% data set. Each of the models was run 10,000 times. Ranks were determined by 50% values and in case of same values, the model with fewer features was ranked higher.

## Data Availability

The data presented in this study are available on request from the corresponding author.

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
