# Peer review of "Association between Self-Reported Prior Night’s Sleep and Single-Task Gait in Healthy, Young Adults: A Study Using Machine Learning"

_sensors, 2022, doi:10.3390/s22197406_

Round 1

Reviewer 1 Report

Thank you for giving me the opportunity to review this interesting manuscript draft.

In this paper, the authors examine the relationship between previous night's sleep duration and single-task gait in healthy young adults. It would be wonderful if this research could lead to injury prevention among young adults, etc.

However, I have some comments for the paper.

If possible, please provide any literature that supports your discussion in lines 278-281.

Is there any literature on the effects of different walking tasks on attention and supraspinous input to gait?

Is there any literature that supports that two-step command in new and clinical setting alter the available resources of the executive control.

Author Response

We believe we have fully addressed the revisions requested. Additionally, we would like to express our appreciation to the reviewer for providing their feedback. We believe that their reviews have made the manuscript significantly stronger. Below are the reviewer’s comments and our responses are in red.

Reviewer #1:

  1. Existing literature can be improved. 

Response: Thank you for the feedback and bringing this to our attention. We added several more citations to the paper. In addition to addressing these concerns in the introduction, we have also added ciations to the discussion section.

Some of the citations added are:

Pellegrino R, Kavakli IH, Goel N, et al. A Novel BHLHE41 Variant is Associated with Short Sleep and Resistance to Sleep Deprivation in Humans. Sleep. 2014;37(8):1327-1336. doi:10.5665/sleep.3924

Li Y, Sahakian BJ, Kang J, et al. The brain structure and genetic mechanisms underlying the nonlinear association between sleep duration, cognition and mental health. Nat Aging. 2022;2(5):425-437. doi:10.1038/s43587-022-00210-2

Reis C, Dias S, Rodrigues AM, et al. Sleep duration, lifestyles and chronic diseases: a cross-sectional population-based study. Sleep Sci. 2018;11(4):217-230. doi:10.5935/1984-0063.20180036

Kokkotis C, Moustakidis S, Tsatalas T, et al. Leveraging explainable machine learning to identify gait biomechanical parameters associated with anterior cruciate ligament injury. Sci Rep. 2022;12(1):6647. doi:10.1038/s41598-022-10666-2

Umemura GS, Pinho JP, Duysens J, Krebs HI, Forner-Cordero A. Sleep deprivation affects gait control. Sci Rep. 2021;11(1):21104. doi:10.1038/s41598-021-00705-9

  1. There is no scientific justification as to why <7 and > 9-hour sleepers are considered poor and the rest as good. 

Response: This is an important matter that the reviewer points out. We based on our dichotomous variable of ‘good’ and ‘poor’ sleepers base on the recommendations from concensus statements fropanels of experts The American Academy of Sleep Medicine (AASM) and Sleep Research Society (SRS)(see Watson et al. below) and the National Sleep Foundation (See Hirshkowitz below). These recommendations are based a review of scientific evidence examining the relationship between sleep duration and health. While these may be appropriate on the population level there is a growing body of literature acknowledging a strong genetic component in terms of individual sleep needs (see Pellegrino et al. below). This literature supports that some individuals may function equally as well to others on less sleep. Sleep extension has been much less studies in the existing literature; However, several recent studies have reported that excessive sleep duration is associated with cognitive decline (See Li below, 2022) and chronic disease, worse quality of life and bad physical function (see Reis below). Considering this information, as well as data visuals mentioned in our preliminary analyses, we felt the best approach for our study was to create the dichotomous variable of ‘good’ and ‘poor’ sleepers. This approach is similar to that taken by other studies investigating differences in individuals meeting, or not meeting, public health recommendations for physical activity (See Totosy de Zepetnek below).  Also, our present study is one of the first to examine the effect of prior night sleep on gait in healthy, young adults. Future work in this area should consider our limitations to improve methodology in subsequent investigations.

To address this comment we have added additional comments to the manuscript in the introduction and limitations to provide a more complete discussion surrounding these issues. We appreciate the feedback from the reviewer.

Watson NF, Badr MS, Belenky G, et al. Recommended Amount of Sleep for a Healthy Adult: A Joint Consensus Statement of the American Academy of Sleep Medicine and Sleep Research Society. J Clin Sleep Med. 2015;11(6):591-592. doi:10.5664/jcsm.4758

Hirshkowitz M, Whiton K, Albert SM, et al. National Sleep Foundation’s updated sleep duration recommendations: final report. Sleep Health. 2015;1(4):233-243. doi:10.1016/j.sleh.2015.10.004

Pellegrino R, Kavakli IH, Goel N, et al. A Novel BHLHE41 Variant is Associated with Short Sleep and Resistance to Sleep Deprivation in Humans. Sleep. 2014;37(8):1327-1336. doi:10.5665/sleep.3924

Li Y, Sahakian BJ, Kang J, et al. The brain structure and genetic mechanisms underlying the nonlinear association between sleep duration, cognition and mental health. Nat Aging. 2022;2(5):425-437. doi:10.1038/s43587-022-00210-2

Reis C, Dias S, Rodrigues AM, et al. Sleep duration, lifestyles and chronic diseases: a cross-sectional population-based study. Sleep Sci. 2018;11(4):217-230. doi:10.5935/1984-0063.20180036

Totosy de Zepetnek JO, Martin J, Cortes N, Caswell S, Boolani A. Influence of grit on lifestyle factors during the COVID-19 pandemic in a sample of adults in the United States. Pers Individ Dif. 2021;175:110705. doi:10.1016/j.paid.2021.110705

  1. For the ML model, what is the total dataset size? 

Response: The total data size was 123 participants. We report this in our methodology section under section 2.3.1

  1. What are the number of epochs that are given for your model? 

Response: We used the default setting for Scikit-learn (Sklearn, Python, https://www.python.com) library to stop training when the cost is converged.

  1. Why did you choose to run the model 10,000 times? Justification is missing. 

Response: Thank you for this querry. We used a Monte Carlo method to simulate the model 10,000 based on the work by Irizarry on using the Monte Carlo method to select random data and running them 10,000 times for life sciences related machine learning models.

Irizarry, R. A. (2019). Introduction to data science: Data analysis and prediction algorithms with R. CRC Press.

Reviewer 2 Report

1. Existing literature can be improved. 

2. There is no scientific justification as to why <7 and > 9-hour sleepers are considered poor and the rest as good. 

3. For the ML model, what is the total dataset size? 

4. What are the number of epochs that are given for your model? 

5. Why did you choose to run the model 10,000 times? Justification is missing. 

Author Response

We believe we have fully addressed the revisions requested. Additionally, we would like to express our appreciation to the reviewer for providing their feedback. We believe that their reviews have made the manuscript significantly stronger. Below are the reviewer’s comments and our responses are in red.

Reviewer #2:

The authors present a study that compared gait performances in subjects after partial sleep deprivation (less than 7 hours of sleep) or increased sleep time (more than 9 hours of sleep).

The paper is well-written and presented. Some comments are below:

- subjects self-reported their sleep duration/quality, whereas the authors could use a sleep monitor to track the time spent sleeping and make the research framework more robust

Response: Thank you for raising this issue and a limitation of the current study. The data presented were part of a larger study with prior night sleep being a secondary focus. As a result during the conceptualized and methodology designed we neglected to incorporate the use of a sleep monitor to strengthen this aspect of the overall study. We hope that our acknowledgement of this limitation (line 323) and recommendation for future research (line 331) to incorporate sleep monitoring advices is implemented with future work in this area.

- the authors should comment on the potential causes of the low classification accuracy

Response: We appreciate the reviewer allowing us the opportunity to identify some reasons why there was low classification accuracy. Based on our data, the ideal model to run would have been to run a machine learning algorithm that is good at predicting parabolic data. However, since the objective of this study was to identify individuals who did not sleep the recommended amount and those who slept the recommended amount a nonlinear parabolic partial differential equation system was not ideal. Therefore, we assume that since individuals who might have fallen right outside of the 7-9 hour range (i.e. 6.9 hours and 9.1 hours) had gait parameters very similar to the ones who slept 7.1 hours or 8.9 hours which would significantly reduce our model accuracy. We have now added this to the limitations of our discussion section.

Minor fixes:

- double parentheses in abstract

Response: Thank you for pointing this out, this has been fixed.

- use "hours" instead of "h" in text (e.g., 7-9 hours)

Response: This has been fixed throughout the manuscript.

- do not specify that the study was approved by the IRB: as human subject research, it must have been approved

Response: While we appreciate this feedback, we disagree with the reviewer as we would like to make the readers aware that all ethical considerations were undertaken for this research.

- do not specify the version of Python and more details on the organization (e.g., location)

Response: We appreciate the reviewers feedback and we have added additional details about the location of the campus from which the participants were recruited.

-  table 1 and caption have an inconsistent alignment with the rest of the paper

Response: We noticed this and was aligned based on the template provided on MDPI website. We are happy to adjust the alignment if needed to but will have to defer to the editing team for how to proceed. 

Reviewer 3 Report

The authors present a study that compared gait performances in subjects after partial sleep deprivation (less than 7 hours of sleep) or increased sleep time (more than 9 hours of sleep).

The paper is well-written and presented. Some comments are below:

- subjects self-reported their sleep duration/quality, whereas the authors could use a sleep monitor to track the time spent sleeping and make the research framework more robust
- the authors should comment on the potential causes of the low classification accuracy

Minor fixes:

- double parentheses in abstract
- use "hours" instead of "h" in text (e.g., 7-9 hours)
- do not specify that the study was approved by the IRB: as human subject research, it must have been approved
- do not specify the version of Python and more details on the organization (e.g., location)
-  table 1 and caption have an inconsistent alignment with the rest of the paper

Author Response

We believe we have fully addressed the revisions requested. Additionally, we would like to express our appreciation to the reviewer for providing their feedback. We believe that their reviews have made the manuscript significantly stronger. Below are the reviewer’s comments and our responses are in red.

Reviewer #3

In this paper, the authors examine the relationship between previous night's sleep duration and single-task gait in healthy young adults. It would be wonderful if this research could lead to injury prevention among young adults, etc.

Response: We appreciate the reviewer’s feedback and agree that this sort of work could be very useful for injury prevention research. Our goal is to apply this work.

However, I have some comments for the paper.

If possible, please provide any literature that supports your discussion in lines 278-281.

Is there any literature on the effects of different walking tasks on attention and supraspinous input to gait?

Response: In our paper we spoke of “supraspinal” input meaning input from above the spinal cord. Kuhn et al studied the impact of external attentional focus on supraspinal input and found that adopting an external attentional focus reduced oxygen consumption in the brain and improved motor outputs during easy to measure, isolated movements.

Kuhn, Y. A., Keller, M., Ruffieux, J., & Taube, W. (2017). Adopting an external focus of attention alters intracortical inhibition within the primary motor cortex. Acta Physiologica, 220(2), 289-299.
https://onlinelibrary.wiley.com/doi/pdf/10.1111/apha.12807

Two other studies found similar results in running:

Schucker, L., Hagemann, N., Strauss, B. & V € olker, K. 2009. € The effect of attentional focus on running economy. J Sports Sci 27, 1241–1248.

Schucker, L., Anheier, W., Hagemann, N., Strauss, B. & € Volker, K. 2013. On the optimal focus of attention for efficient running at high intensity. Sport Exerc Perform Psychol 2, 207–219.

We have now added these citations to our study.

Is there any literature that supports that two-step command in new and clinical setting alter the available resources of the executive control.

Response:There is literature that supports our assertions. Coppin et al looked specifically at the effects of different dual tasks on gait. They found that certain tasks, like walking with obstacles had greater dual task costs than other tasks like walking and talking. These dual task costs were higher in individuals with executive function deficits suggesting that they affect the available resources of the executive control specifically. We have now added this citation to our study.

Coppin, A. K., Shumway-Cook, A., Saczynski, J. S., Patel, K. V., Ble, A., Ferrucci, L., & Guralnik, J. M. (2006). Association of executive function and performance of dual-task physical tests among older adults: analyses from the InChianti study. Age and ageing35(6), 619-624.

Round 2

Reviewer 2 Report

Thank you for considering my suggestions.